

# Genetic determinants of COVID-19 severity and mortality: *ACE1* Alu 287 bp polymorphism and *ACE1*, *ACE2*, *TMPRSS2* expression in hospitalized patients

João Locke Ferreira de Araújo[1,2], Átila Duque Rossi[3], Jessica Maciel de Almeida[3], Hugo José Alves[1], Isabela de Carvalho Leitão[4], Renata Eliane de Ávila[5], Anna Carla Pinto Castiñeiras[4], Jéssica da Silva Oliveira[6], Rafael Mello Galliez[4], Marlon Daniel Lima Tonini[6], Débora Souza Faffe[4], Shana Priscila Coutinho Barroso Barroso[6,7], Gustavo Gomes Resende[8], Cássia Cristina Alves Gonçalves[3,4], Terezinha Marta Pereira Pinto Castiñeiras[4], Amilcar Tanuri[3], Mauro Martins Teixeira[9], Renato Santana Aguiar[1,10], Cynthia Chester Cardoso[3] and Renan Pedra de Souza[1]

[1] Departamento de genética, ecologia e evolução, Laboratório de biologia integrativa, Universidade Federal de Minas Gerais, Belo Horizonte, Minas Gerais, Brazil
[2] Departamento de biorregulação, Laboratório de imunofarmacologia e biologia molecular, Universidade Federal da Bahia, Salvador, BA, Brazil
[3] Departamento de genética, Laboratório de Virologia Molecular, Universidade Federal do Rio de Janeiro, Rio de Janeiro, Rio de Janeiro, Brazil
[4] Núcleo de Enfrentamento e Estudos de Doenças Infecciosas Emergentes e Reemergentes (NEEDIER), Universidade Federal do Rio de Janeiro, Rio de Janeiro, Rio de Janeiro, Brazil
[5] Hospital Eduardo de Menezes, Belo Horizonte, Minas Gerais, Brazil
[6] Marinha do Brasil, Instituto de Pesquisas Biomédicas, Hospital Naval Marcilio Dias, Rio de Janeiro, Rio de Janeiro, Brazil
[7] Clínica RioVet, Rio de Janeiro, Rio de Janeiro, Brazil
[8] Hospital das Clínicas, (HC-UFMG/EBSERH), Belo Horizonte, MG, Brazil, Universidade Federal de Minas Gerais, Belo Horizonte, Minas Gerais, Brazil
[9] Departamento de Bioquímica e Imunologia, Instituto de Ciências Biológicas, Universidade Federal de Minas Gerais, Belo Horizonte, Minas Gerais, Brazil
[10] Instituto D'OR de Pesquisa e Ensino, Rio de Janeiro, Rio de Janeiro, Brazil

Corresponding authors
João Locke Ferreira de Araújo,
joaolocke.bio@gmail.com
Renan Pedra de Souza,
renanpedra@gmail.com,
renanrps@ufmg.br

## ABSTRACT

**Background**. The angiotensin-converting enzyme 2 (ACE2) and the transmembrane serine protease 2 (TMPRSS2) are central human molecules in the SARS-CoV-2 virus-host interaction. Evidence indicates that *ACE1* may influence *ACE2* expression. This study aims to determine whether ACE1, ACE2, and TMPRSS2 mRNA expression levels, along with the ACE1 Alu 287 bp polymorphism (rs4646994), contribute to the severity and mortality of COVID-19.

**Methods**. Swabs were collected in two Brazilian cities in 2020: Belo Horizonte ($n = 134$) and Rio de Janeiro ($n = 41$). A swab of mild patients in Rio de Janeiro who were not hospitalized ($n = 172$) was also collected. All analyzed biological material was obtained from residual diagnostic samples in 2020, prior to the emergence of SARS-CoV-2 variants of concern. *ACE1*, *ACE2*, *TMPRSS2*, and *B2M* (reference gene) expression
levels were evaluated in 40 cycles of quantitative PCR. *ACE1* Alu 287 bp polymorphism was genotyped using the FastStart Universal SYBR Green Master kit.

**Results**. The median age differed between clinical sites ($p = 0.016$), but no difference in median days of hospitalization was observed ($p = 0.329$). Age was associated with severity ($p = 0.014$) and mortality ($p = 0.014$) in the Belo Horizonte cohort. No alteration in *ACE1*, *ACE2* and *TMPRSS2* expression was associated with severity or mortality. *ACE1* polymorphism rs4646994 did not influence the likelihood of either outcome. A meta-analysis including available data from the literature showed significant effects: the D-allele conferred risk (OR = 1.39; 95% CI [1.12–1.72]).

## INTRODUCTION

Coronavirus 2019 disease (COVID-19) is caused by a virus of the *Coronaviridae* family, known as the Severe Acute Respiratory Syndrome Coronavirus 2 (SARS-CoV-2). The clinical manifestation of COVID-19 can be highly heterogeneous with patients ranging from asymptomatic to severe cases. Various clinical, genetic, and epidemiological factors have been linked to COVID-19 severity worldwide (*Marcolino et al., 2021*; *De Araújo et al., 2022*; *De Araújo et al., 2023*; *Brizzi et al., 2022*). The degree of severity of COVID-19, or vulnerability to SARS-CoV-2, depends on many factors, including genetic polymorphisms, which are studied in the following: transmembrane protease serine 2 (TMPRSS2), tumor necrosis factor-alpha (TNF-α), interferon-gamma (IFN-γ), and angiotensin-converting enzyme II (*De Araújo et al., 2022*; *Akbari et al., 2022*; *Zhang et al., 2022*).

The angiotensin-converting enzyme 2 (ACE2) and TMPRSS2 are central human molecules in the virus-host interaction (*Muus et al., 2021*). The spike viral protein interacts with the ACE2 receptor, and TMPRSS2 cleaves the spike protein's receptor binding domain (RBD) exposing a fusion peptide (*Hoffmann et al., 2020*). Preliminary studies have explored the association between *ACE2* and *TMPRSS2* gene expression and their polymorphisms with COVID-19 outcomes (*Rossi et al., 2021*; *COVID-19 Host Genetics Initiative, 2022*; *Taglauer et al., 2022*; *Saengsiwaritt et al., 2022*). Significant expression alterations were found in subjects presenting respiratory distress (*Rossi et al., 2021*).

The angiotensin-converting enzyme 1 (ACE1) catalyzes the conversion of angiotensin I to angiotensin II, an ACE2 substrate. Evidence indicates that *ACE1* may influence *ACE2* expression (*Hamdi & Castellon, 2004*). An *ACE1* 287bp insertion/deletion polymorphism (rs4646994) has been associated with increased *ACE1* enzyme activity in homozygous individuals for the deletion allele (D/D) (*Suehiro et al., 2004*). A recent meta-analysis showed a 45% increase in the chance of severe COVID-19 manifestation in *ACE1* deletion carriers, although no effect on susceptibility was found (*De Araújo et al., 2022*).

Identifying biomarkers associated with COVID-19 outcomes will help clarify its pathophysiology and improve prognosis. Proteins related to virus-host interaction are strong candidates for biomarkers. Therefore, we evaluated whether *ACE1*, *ACE2*, and

*TMPRSS2* gene expression and *ACE1* polymorphism (Alu 287 bp) would contribute to the need for mechanical ventilation and chance of death in a cohort of hospitalized COVID-19 patients in Brazil.

## MATERIALS AND METHODS

Portions of this text were previously published as part of a thesis (http://hdl.handle.net/1843/55939). Enrolled subjects were inpatients from two Brazilian hospitals: Hospital Naval Marcilio Dias (HNMD) in Rio de Janeiro ($n = 41$) and Eduardo de Menezes (HEM) in Belo Horizonte ($n = 134$). Additionally, 172 patients with mild symptoms collected at the Centro de Triagem e Diagnóstico de COVID-19 from the Universidade Federal do Rio de Janeiro (UFRJ) were included in a second cohort in Rio de Janeiro for genetic association studies with the Alu 287 bp (rs4646994) polymorphism. All biological materials analyzed were obtained from residual diagnostic samples collected in 2020, prior to the emergence of SARS-CoV-2 variants of concern. Samples from Rio de Janeiro consisted of nasopharyngeal swabs, while samples from Belo Horizonte included nasopharyngeal swabs ($n = 102$) and bronchoalveolar lavage ($n = 32$). The study adhered to the Declaration of Helsinki and was approved by the Ethics Committees.

Participant information was collected from medical records or from forms completed by volunteers at UFRJ. All participants provided written informed consent approved by the institutional ethics review boards from UFRJ, HMND, HEM, and Universidade Federal de Minas Gerais (protocols 30161620.0.0000.5257, 32382820.3.0000.5256, 32224420.3.0000.0008, and 31462820.3.0000.5149, respectively). For patients unable to provide consent due to hospitalization, consent was obtained from a legal guardian (*Rossi et al., 2021*).

Biomarker effects were explored in two outcomes: the need for mechanical ventilation during hospitalization (using both samples) and mortality (using the Belo Horizonte sample, as no deaths were recorded in the Rio de Janeiro cohort). Mechanical ventilation was considered a severity criterion for the hospitalized patient sample. Additionally, severity in the Rio de Janeiro cohort was assessed by evaluating the likelihood of hospitalization. All molecular experiments were conducted blinded to outcome information.

Samples were collected in viral transport medium and stored at $-80\,°C$ until extraction. RNA and DNA extractions were performed using the Quick-RNA Viral kit (Zymo Research, Irvine, CA, USA), following the manufacturer's instructions and standardized laboratory protocols. cDNA synthesis was carried out using the High-capacity cDNA Reverse Transcription Kit (Thermo Fisher Scientific, Waltham, MA, USA) as per the manufacturer's instructions.

*ACE1*, *ACE2*, *TMPRSS2*, and *B2M* (reference gene) expression levels were measured through quantitative PCR using Integrated DNA Technologies (Coralville, IA, USA) exon-exon junction probes (Hs.PT.58.19167084, Hs.PT.58.27645939, HS.PT.58.39738666, and Hs.PT.39a.22214847). The $\Delta Ct$ values were calculated by subtracting the cycle threshold (Ct) of the gene of interest from the Ct of B2M. All samples that amplified the reference gene were included in the analysis. This gene expression assay was standardized and has been described previously (*Braga-Paz et al., 2022*).
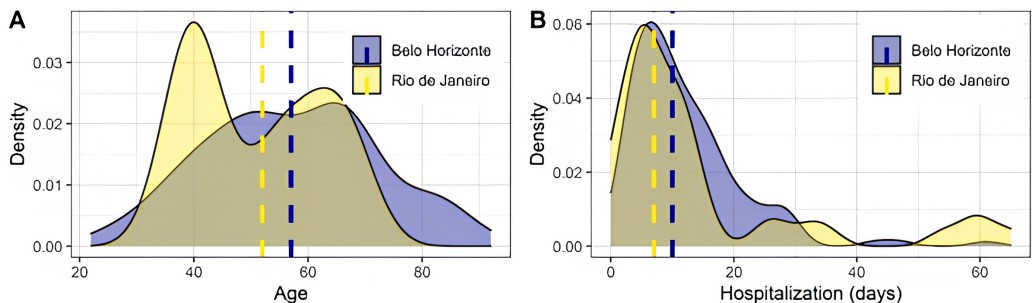

**Figure 1** **Distribution of age and hospitalization days across the Belo Horizonte and Rio de Janeiro samples.** Dashed lines represent medians. The difference in median age was significant ($p = 0.016$), assessed using the Mann–Whitney test. No significant difference was found in the median number of hospitalization days ($p = 0.329$), also assessed using the Mann–Whitney test.

Gene expression analyses were performed exclusively on samples extracted from nasopharyngeal tissue (Rio de Janeiro ($n = 41$) and Belo Horizonte ($n = 102$)). Samples without amplification of the target gene were assigned a Ct value of 40 (minimum expression level). Only nasopharyngeal swab samples were included in the gene expression analysis.

ACE1 Alu 287 bp polymorphism was genotyped using the FastStart Universal SYBR Green Master kit (Promega, WI, USA), following the method adapted from *Evans et al. (1994)* and previously described by *Braga-Paz et al. (2022)*. The reaction used three primers: 5′CATCCTTTCTCCCATTTCTC3′ (Primer1, Forward), 5′TGGGATTACAGGCGTGATACAG3′ (Primer 2, Forward, internal), and 5′ATTTCAGAGCTGGAATAAAATT3′ (Primer 3, Reverse). Primer stocks were resuspended at 100 μM and diluted to a 10 μM working solution. Final primer concentrations were 20 picomoles for Primers 1 and 3, and 40 picomoles for Primer 2. Fragment sizes of 65 bp (insertion) and 84 bp (deletion) were visualized on a 3% agarose gel. To ensure genotyping quality, 10% of the samples were randomly re-genotyped, showing 100% agreement. The genotyping protocol has been described previously (*Braga-Paz et al., 2022*).

Statistical analyses were conducted using the R software environment (version 4.1.2). Data normality was assessed using the Shapiro–Wilk test. Clinical data were compared using the Mann–Whitney and Fisher's Exact tests. Deviations from Hardy-Weinberg equilibrium were evaluated in cases and controls using Pearson's chi-squared test within the SNPassoc package (*González et al., 2022*), with no violations observed ($p > 0.05$ for all samples). Median ΔCt differences were assessed using the Mann–Whitney test. Genetic associations with outcomes were analyzed using Pearson's chi-squared or Fisher's Exact tests, respecting the assumptions of each test. Figure 1 was created using the ggplot2 package (*Wickham, 2024*). Combined polymorphism effects were determined through meta-analysis using the Mantel–Haenszel weighted means method under the fixed-effect model implemented in the metabin function (*Schwarzer, Carpenter & Rücker, 2015*). A significance level of 5% was set.

**Table 1  Comparison of clinical and epidemiological data between clinical sites.** Data are presented as absolute and relative frequencies.

| Variable | Rio de Janeiro (swab), $n = 41$ | Belo Horizonte (swab+BAL), $n = 134$ | $p$-value[a] | Belo Horizonte (swab only), $n = 102$ | $p$-value[b] |
|---|---|---|---|---|---|
| Sample from swab - $n$ (%) | 41 (100%) | 102 (76%) | – | 102 (100%) | 0.999 |
| Female - $n$ (%) | 26 (63%) | 66 (49%) | 0.112 | 54 (53%) | 0.254 |
| Comorbidity - $n$ (%) | 27 (66%) | 93 (69%) | 0.668 | 69 (68%) | 0.836 |
| Chronic medication use - $n$ (%) | 23 (79%) | 95 (71%) | 0.358 | 69 (68%) | 0.225 |
| Fever - $n$ (%) | 33 (80%) | 104 (78%) | 0.754 | 80 (79%) | 0.864 |
| Chills - $n$ (%) | 2 (4.9%) | 5 (3.7%) | 0.667 | 4 (3.9%) | 0.999 |
| Cough - $n$ (%) | 31 (76%) | 111 (83%) | 0.301 | 87 (85%) | 0.168 |
| Sneezing - $n$ (%) | 5 (12%) | 15 (12%) | 0.999 | 9 (9.2%) | 0.554 |
| Dyspnea - $n$ (%) | 34 (83%) | 114 (85%) | 0.739 | 84 (82%) | 0.935 |
| Coryza - $n$ (%) | 7 (17%) | 42 (31%) | 0.075 | 32 (31%) | 0.083 |
| Headache - $n$ (%) | 11 (27%) | 42 (31%) | 0.582 | 36 (35%) | 0.330 |
| Adynamia - $n$ (%) | 4 (9.8%) | 89 (66%) | **<0.001** | 58 (57%) | **<0.001** |
| Nausea - $n$ (%) | 4 (9.8%) | 18 (13%) | 0.534 | 16 (16%) | 0.355 |
| Vomit - $n$ (%) | 2 (4.9%) | 24 (18%) | **0.039** | 19 (19%) | **0.034** |
| Diarrhea - $n$ (%) | 7 (17%) | 33 (25%) | 0.313 | 25 (25%) | 0.335 |
| Myalgia - $n$ (%) | 19 (46%) | 54 (40%) | 0.492 | 50 (49%) | 0.772 |
| Anosmia - $n$ (%) | 7 (17%) | 20 (15%) | 0.739 | 18 (18%) | 0.935 |
| Ageusia - $n$ (%) | 5 (12%) | 11 (8.2%) | 0.535 | 10 (9.8%) | 0.764 |
| Fatigue - $n$ (%) | 11 (27%) | 27 (20%) | 0.364 | 18 (18%) | 0.217 |
| Intensive care unit - $n$ (%) | 10 (26%) | 76 (57%) | **<0.001** | 48 (47%) | **0.027** |
| Respiratory support - any - $n$ (%) | 38 (93%) | 133 (99%) | **0.041** | 101 (99%) | 0.071 |
| Respiratory support - catheter - $n$ (%) | 22 (54%) | 109 (81%) | **<0.001** | 92 (90%) | **<0.001** |
| Respiratory support - mask - $n$ (%) | 7 (17%) | 63 (47%) | **<0.001** | 41 (40%) | **0.008** |
| Respiratory support - mechanical ventilation - $n$ (%) | 9 (22%) | 59 (48%) | **0.004** | 29 (32%) | 0.259 |

**Notes.**

$n$, sample size; BAL, bronchoalveolar lavage; Swab, nasal swab.

[a] Association $p$-values computed using patients from Rio de Janeiro (swab) and Belo Horizonte (swab + BAL).

[b] Association $p$-values computed using patients from Rio de Janeiro (swab) and Belo Horizonte (swab). Statistical significance was assessed using Fisher's exact test.

## RESULTS

Clinical data were compared between recruitment sites. A difference in median age was observed ($p = 0.016$), with no difference in median days of hospitalization ($p = 0.329$) (Fig. 1). Most evaluated symptoms were homogeneously distributed, except for adynamia and vomiting (Table 1). Clinical outcomes also showed significance between sites, with Belo Horizonte presenting increased severity, as shown by the association of admission to the intensive care unit and respiratory support type (Table 1). It was observed that 34 deaths occurred in the Belo Horizonte cohort (25% of the sample). In contrast, no patients died in the Rio de Janeiro cohort.

Median *ACE1*, *ACE2,* and *TMPRSS2* gene expression did not significantly differ according to both investigated outcomes (the need for mechanical ventilation and death) in hospitalized patients (Table 2). Furthermore, the median ratio between *TMPRSS2* and

**Table 2  Evaluation of ACE1, ACE2, and TMPRSS2 expression levels in COVID-19 outcomes.** No significant expression differences were found. Experiments were conducted on nasopharyngeal tissue samples.

| Variable | Need for mechanical ventilation (Rio de Janeiro); $n = 41$ | | | Need for mechanical ventilation (Belo Horizonte); n=102 | | | Death (Belo Horizonte); $n = 102$ | | |
|---|---|---|---|---|---|---|---|---|---|
| | No, $n = 32$ | Yes, $n = 9$ | *p*-value | No, $n = 63$ | Yes, $n = 29$ | *p*-value | No, $n = 88$ | Yes, $n = 14$ | *p*-value |
| Age - median (interquartile range) missing data | 41 (40, 59) 2 | 55 (52, 63) | 0.291 | 54 (44, 65) 0 | 54 (48, 68) 0 | 0.215 | 52 (44, 63) 0 | 68 (64, 82) 0 | **<0.001** |
| ACE1 delta Ct - median (interquartile range) missing data | Not available | Not available | Not available | 11.3 (8.4, 13.4) 12 | 10.3 (7.7, 13.0) 8 | 0.552 | 11.3 (8.5, 13.4) 17 | 8.8 (6.8, 11.1) 3 | 0.226 |
| ACE2 delta Ct - median (interquartile range) missing data | 6.40 (4.97, 7.83) 0 | 8.65 (5.36, 8.72) 0 | 0.128 | 15.1 (12.2, 17.7) 11 | 12.9 (10.8, 15.9) 3 | 0.192 | 13.5 (10.7, 17.5) 13 | 14.4 (12.4, 15.4) 1 | 0.888 |
| TMPRSS2 delta Ct - median (interquartile range) missing data | 4.57 (3.72, 5.65) 0 | 4.87 (4.18, 9.38) 0 | 0.206 | 9.0 (5.3, 11.8) 11 | 8.4 (4.5, 13.8) 3 | 0.845 | 8.4 (4.9, 12.1) 13 | 8.4 (5.8, 10.3) 1 | 0.925 |
| ACE2/TMPRSS2 delta Ct ratio - median (interquartile range) missing data | 1.26 (1.16, 1.64) | 1.37 (1.19, 1.51) | 0.938 | 1.58 (1.00, 2.33) 11 | 1.40 (1.00, 2.46) 3 | 0.582 | .54 (1.00, 2.58) 13 | 1.43 (1.00, 1.92) 1 | 0.972 |

**Notes.**

n, sample size.

Statistical significance was assessed using the Mann-Whitney test.

*ACE2* expression did not show an effect. As expected, increased median age was found among subjects who died compared to those who survived.

No association was found between *ACE1* Alu 287 bp polymorphism and the need for mechanical ventilation or death (Table 3). When testing hospitalized *versus* non-hospitalized patients from Rio de Janeiro, there was a difference in age ($p < 0.001$) although no association was observed for Alu 287 bp polymorphism either (Table 4).

All models were adjusted for age, taking into account their individual significance. However, no analysis demonstrated any alteration.

Combined effects from both samples on the need for mechanical ventilation also did not reach significance: pooled odds-ratio for D-allele dominance was 1.12 (95% confidence interval: 0.58–2.18) (Supplemental Information 1). Since the number of subjects varied from the expression analysis, we reevaluated the age effect and observed a significant median difference in the Belo Horizonte sample for both outcomes.

We carried out a literature search in the Pubmed database, complementary to our previous work (*De Araújo et al., 2022*) to evaluate the combined effects. The review followed the parameters recommended by the Preferred Reporting Items for Systematic Reviews and meta-analysis (PRISMA), following the steps of identification, screening, and eligibility. A search strategy was devised following a Boolean logic containing terms related to COVID-19

de Araújo et al. (2025), *PeerJ*, DOI 10.7717/peerj.18508

**Table 3 Association of ACE1 Alu 287 bp polymorphism with the need for mechanical ventilation or death in patients.** No significant association was observed. Differences between the sample size and genotype counts are due to failed genotyping reactions.

| Variable | | Need for mechanical ventilation (Rio de Janeiro) | | | Need for mechanical ventilation (Belo Horizonte) | | | Death (Belo Horizonte) | | |
|---|---|---|---|---|---|---|---|---|---|---|
| | | No, *n* = 32 | Yes, *n* = 9 | *p*-value | No, *n* = 65 | Yes, *n* = 59 | *p*-value | No, *n* = 100 | Yes, *n* = 34 | *p*-value |
| Age - median (interquatile range) | | 41 (40, 59) | 55 (52, 63) | 0.291 | 54 (44, 65) | 63 (48, 69) | **0.014** | 54 (44, 65) | 67 (59, 80) | **<0.001** |
| | D/D - *n* (%) | 10 (31%) | 2 (22%) | | 23 (36%) | 24 (42%) | | 39 (39%) | 16 (50%) | |
| Co-dominance | D/I - *n* (%) | 16 (50%) | 4 (44%) | 0.698 | 27 (42%) | 23 (40%) | 0.739 | 40 (40%) | 11 (34%) | 0.566 |
| | I/I - *n* (%) | 6 (19%) | 3 (33%) | | 14 (22%) | 10 (18%) | | 20 (20%) | 5 (16%) | |
| I-allele dominance | DD - *n* (%) | 10 (31%) | 2 (22%) | 0.702 | 23 (36%) | 24 (42%) | 0.487 | 39 (39%) | 16 (50%) | 0.291 |
| | II + DI - *n* (%) | 22 (69%) | 7 (78%) | | 41 (64%) | 33 (58%) | | 60 (61%) | 16 (50%) | |
| D-allele dominance | DD + DI - *n* (%) | 26 (81%) | 6 (67%) | 0.384 | 50 (78%) | 47 (82%) | 0.551 | 79 (80%) | 27 (84%) | 0.567 |
| | I - *n* (%) | 6 (19%) | 3 (33%) | | 14 (22%) | 10 (18%) | | 20 (20%) | 5 (16%) | |

**Notes.**

n, sample size.

Statistical significance was assessed using Fisher's exact test and the Mann-Whitney test.

**Table 4** **Association of ACE1 Alu 287 bp polymorphism with hospitalization in the Rio de Janeiro sample.** Differences between the sample size and genotype counts are due to failed genotyping reactions.

| Variable | | Non-Hospitalized n = 172 | Hospitalized n = 41 | p-value |
|---|---|---|---|---|
| | Age - median (interquartile range) | 39 (30,44) | 52 (40,62) | **<0.001** |
| | D/D - n (%) | 54 (31.8%) | 12 (29.3%) | |
| Co-dominance | D/I - n (%) | 84 (49.4%) | 20 (48.8%) | 0.891 |
| | I/I - n (%) | 32 (18.8%) | 9 (21.9%) | |
| | DD - n (%) | 54 (31.8%) | 12 (29.3%) | |
| I-allele dominance | II + DI - n (%) | 116 (68.2%) | 29 (70.7%) | 0.756 |
| | DD + DI - n (%) | 138 (81.2%) | 32 (77.1%) | |
| D-allele dominance | I - n (%) | 32 (18.8%) | 9 (21.9%) | 0.653 |

Notes.
n, sample size.
Statistical significance was assessed using Fisher's exact test and the Mann-Whitney test

and the pathogen, genetic association studies, and the *ACE1* gene. Search was performed on November 10, 2022. For chance of death, the meta-analysis was performed with two more studies (*Mir et al., 2021*; *Möhlendick et al., 2021*), in which we also did not observe significance: pooled odds-ratio for D-allele dominance was 1.48 (95% confidence interval: 0.38–5.81) (Supplementary Material S2).

We also checked the combined effect of the Rio de Janeiro cohort comparing mild patients with those who required hospitalization with the literature. The meta-analysis was conducted with seven additional studies extracted from the literature (*Gunal et al., 2021*; *Kouhpayeh et al., 2021*; *Saad et al., 2021*; *Verma et al., 2021*; *Gong et al., 2022*; *Martínez-Gómez et al., 2022*; *Mahmood et al., 2022*). We observed significance: pooled odds-ratio for D-allele dominance was 1.39 (95% CI [1.12–1.72]) (Fig. 2).

## DISCUSSION

Molecular signatures associated with COVID-19-related outcomes have been extensively investigated during the pandemic. Molecules related to the immune response have been, by far, the most studied. Among the most significant results, an association was reported between circulating interleukin-6 and COVID-19 severity in a meta-analysis combining 15 original studies (*Zawawi et al., 2021*). Proteins associated with virus-host interaction can also be promising candidates for biomarker studies.

*ACE2* and *TMPRSS2* expressions have been explored due to their central role in the cell entry mechanisms. Higher ACE2 protein levels were found in post-mortem lung samples of patients who died of severe COVID-19 suggesting a pathobiological role in disease severity (*Gheware et al., 2022*). *TMPRSS2/ACE2* expression ratio was associated with respiratory distress (*Rossi et al., 2021*). Moreover, age-dependent *ACE2* expression in the nasal epithelium have been related to lower infection susceptibility and mortality in children (*Bunyavanich, Do & Vicencio, 2020*). However, a recent study did not find differences between infants and adults assessing ACE2 immunofluorescence staining and protein levels (*Zhu et al., 2022*). We report no significant association between *ACE2* and

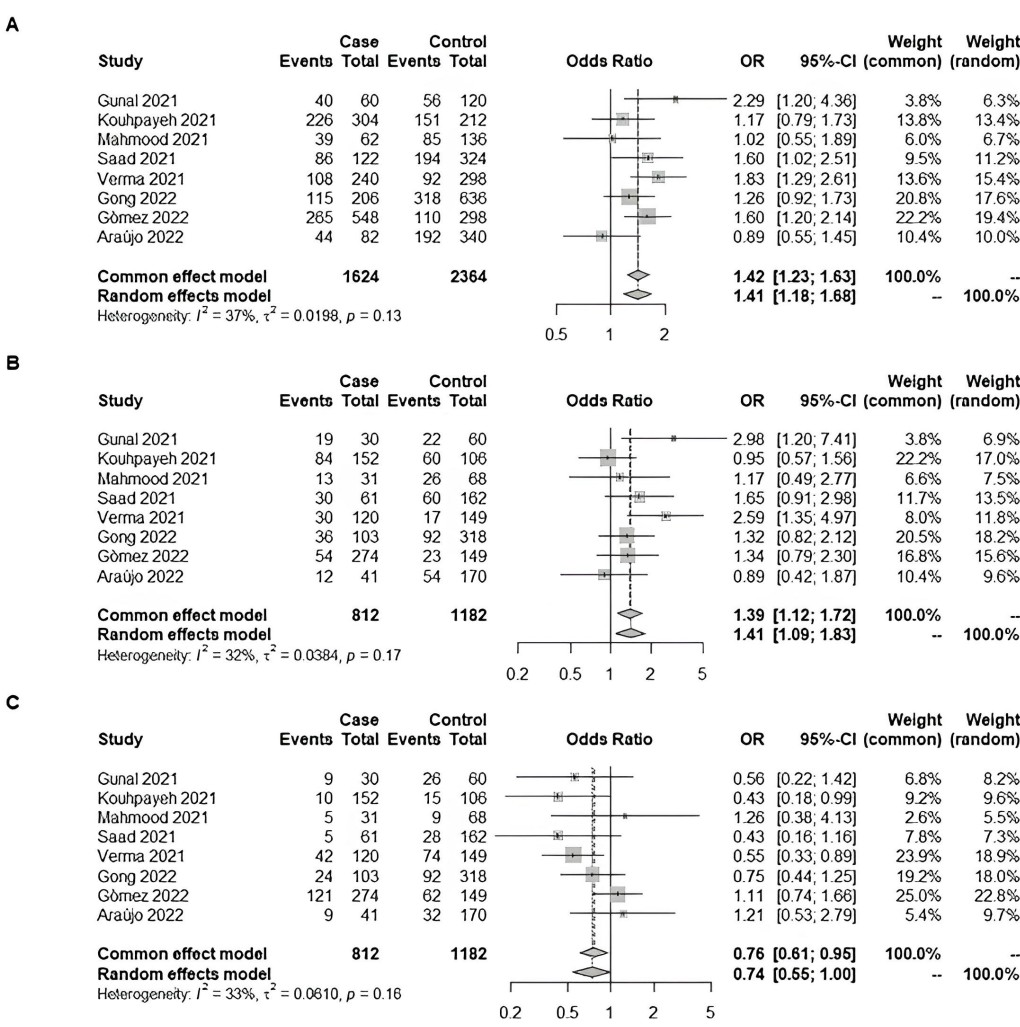

**Figure 2** **Forest plot illustrating the association of ACE1 rs4646994 (Alu 287 bp) with COVID-19 severity (Non-Hospitalized *vs.* Hospitalized).** The effect size from our original study was combined with seven additional studies from the literature using a meta-analysis with the Mantel–Haenszel weighted means method under a fixed-effect model. Significant allelic and genotypic effects were observed. (A) D-allele model: The D-allele was associated with an increased risk of severe COVID-19. (B) D recessive model: D/D genotype carriers had increased odds of severe COVID-19 compared with D/I and I/I carriers combined. (C) I recessive model: I/I genotype carriers had decreased odds of severe COVID-19 compared with D/I and D/D carriers combined Studies: *Gunal et al., 2021*; *Kouhpayeh et al., 2021*; *Mahmood et al., 2022*; *Saad et al., 2021*; *Verma et al., 2021*; *Gong et al., 2022*; *Martínez-Gómez et al., 2022*; *De Araújo et al., 2022*; *De Araújo et al., 2023*.

*TMPRSS2* gene expression and the need for mechanical ventilation or death. Similarly, no *ACE2* expression differences were found between those admitted to the intensive care unit and patients who were not (*Akbari et al., 2020*).

ACE1 also seems to be a good biomarker candidate, although not directly related to viral cell entry. ACE1/ACE2 balance has been hypothesized to contribute to clinical phenotypes relevant to COVID-19 (*Brosnihan, Neves & Chappell, 2005*; *Mizuiri et al.,*

*2008*). ACE1 inhibitors were associated with a significantly reduced risk of hospital admission during COVID-19 in a cohort study including 8.3 million people (*Hippisley-Cox et al., 2020*). We did not find altered *ACE1* expression, although a previous study reported that *ACE1* expression was significantly higher in COVID-19 intensive care unit patients (*Akbari et al., 2022*). Similarly, no association between *ACE1* Alu 287 bp polymorphism and COVID-19 severity was achieved. Although our initial analysis of the ACE1 Alu 287 bp polymorphism did not reveal a significant association with COVID-19 severity, a subsequent meta-analysis that combined our data with seven other studies from the literature did identify a significant association between the D-allele of the ACE1 Alu 287 bp polymorphism and an increased risk of severe COVID-19. This finding suggests that, while individual studies may lack sufficient power, pooling data across multiple studies can uncover important genetic associations with clinical outcomes in COVID-19.

Our report presents limitations. First, replications are warranted because the study may be underpowered to detect minor effects. Second, we could not evaluate the viral diversity impact since samples were collected before describing the variants of concern that substantially changed COVID-19 severity (*Telenti, Hodcroft & Robertson, 2022*). Another relevant factor that could not be explored was the vaccination status. Therefore, additional investigations in larger samples from diverse ethnic backgrounds assessing multiple candidate genes are crucial to understanding COVID-19 prognosis due to its multifactorial structure.

## CONCLUSIONS

Our analysis found no significant association between ACE2 and TMPRSS2 expression and the need for mechanical ventilation or death. Although the ACE1 gene has been considered a promising candidate, we found no significant changes in its expression or in polymorphisms associated with COVID-19 severity, despite observing an association between the rs4646994 polymorphism and hospitalization in the meta-analysis. Considering these results, we emphasize the need for further studies to confirm our findings and explore other possible associations, particularly with respect to viral diversity and patients' vaccination status.

### Funding

We received support from Rede Corona-ômica BR MCTI/FINEP affiliated with RedeVírus/MCTI (01.20.0029.000462/20 404096/2020-4; 1227/21 01.22.0074.00); Conselho Nacional de Desenvolvimento Científico e Tecnológico - CNPq (315592/2021-4); Financiadora de Estudos e Projetos - FINEP (0494/20 01.20.0026.00; 1228/21 01.22.0082.00; 1139/20 01.20.0076.00); Coordenação de Aperfeiçoamento de Pessoal de Nível Superior - CAPES (Finance Code 001); Fundação de Apoio à Pesquisa do Rio de Janeiro - FAPERJ (E-26/210.658/2021). The funders had no role in study design, data collection and analysis, decision to publish, or preparation of the manuscript.

## Grant Disclosures

The following grant information was disclosed by the authors:

Rede Corona-ômica BR MCTI/FINEP affiliated with RedeVírus/MCTI: 01.20.0029.000462/20404096/2020-4, 1227/21 01.22.0074.00.

Conselho Nacional de Desenvolvimento Científico e Tecnológico - CNPq: 01.20.0026.00, 1228/21 01.22.0082.00, 1139/20 01.20.0076.00.

Coordenação de Aperfeiçoamento de Pessoal de Nível Superior - CAPES: 315592/2021-4.

Financiadora de Estudos e Projetos - FINEP: 0494/2001.20.0026.00, 1228/21 01.22.0082.00, 1139/20 01.20.0076.00.

Coordenação de Aperfeiçoamento de Pessoal de Nível Superior - CAPES.

Fundação de Apoio à Pesquisa do Rio de Janeiro - FAPERJ: E-26/210.658/2021.

## Competing Interests

Shana Priscila Coutinho Barroso is a researcher at BioVet, funded by the Foundation for Research Support of the State of Rio de Janeiro –FAPERJ. Renan Pedra de Souza is an Academic Editor for PeerJ.

## Author Contributions

- João Locke Ferreira de Araújo conceived and designed the experiments, performed the experiments, analyzed the data, prepared figures and/or tables, authored or reviewed drafts of the article, and approved the final draft.
- Átila Duque Rossi conceived and designed the experiments, performed the experiments, prepared figures and/or tables, authored or reviewed drafts of the article, and approved the final draft.
- Jessica Maciel de Almeida conceived and designed the experiments, performed the experiments, authored or reviewed drafts of the article, and approved the final draft.
- Hugo José Alves conceived and designed the experiments, performed the experiments, authored or reviewed drafts of the article, and approved the final draft.
- Isabela de Carvalho Leitão conceived and designed the experiments, authored or reviewed drafts of the article, and approved the final draft.
- Renata Eliane de Ávila conceived and designed the experiments, authored or reviewed drafts of the article, and approved the final draft.
- Anna Carla Pinto Castiñeiras conceived and designed the experiments, authored or reviewed drafts of the article, and approved the final draft.
- Jéssica da Silva Oliveira conceived and designed the experiments, authored or reviewed drafts of the article, and approved the final draft.
- Rafael Mello Galliez conceived and designed the experiments, authored or reviewed drafts of the article, and approved the final draft.
- Marlon Daniel Lima Tonini conceived and designed the experiments, authored or reviewed drafts of the article, and approved the final draft.
- Débora Souza Faffe conceived and designed the experiments, authored or reviewed drafts of the article, and approved the final draft.
- Shana Priscila Coutinho Barroso Barroso conceived and designed the experiments, authored or reviewed drafts of the article, and approved the final draft.

- Gustavo Gomes Resende conceived and designed the experiments, authored or reviewed drafts of the article, and approved the final draft.
- Cássia Cristina Alves Gonçalves conceived and designed the experiments, authored or reviewed drafts of the article, and approved the final draft.
- Terezinha Marta Pereira Pinto Castiñeiras conceived and designed the experiments, authored or reviewed drafts of the article, and approved the final draft.
- Amilcar Tanuri conceived and designed the experiments, authored or reviewed drafts of the article, and approved the final draft.
- Mauro Martins Teixeira conceived and designed the experiments, authored or reviewed drafts of the article, and approved the final draft.
- Renato Santana Aguiar conceived and designed the experiments, authored or reviewed drafts of the article, and approved the final draft.
- Cynthia Chester Cardoso conceived and designed the experiments, prepared figures and/or tables, authored or reviewed drafts of the article, and approved the final draft.
- Renan Pedra de Souza conceived and designed the experiments, analyzed the data, prepared figures and/or tables, authored or reviewed drafts of the article, and approved the final draft.

## Human Ethics

The following information was supplied relating to ethical approvals (*i.e.*, approving body and any reference numbers):

Ethics Committee (protocols 30161620.0.0000.5257, 32382820.3.0000.5256, 32224420.3.0000.0008, 31462820.3.0000.5149)

## Data Availability

The raw data is available in the Supplemental File.

## Supplemental Information

Supplemental information for this article can be found online at http://dx.doi.org/10.7717/peerj.18508#supplemental-information.

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
