# Peer review of "Genetic determinants of COVID-19 severity and mortality: ACE1 Alu 287 bp polymorphism and ACE1, ACE2, TMPRSS2 expression in hospitalized patients"

_PeerJ, doi:10.7717/peerj.18508_

## Round 0.1 · original submission · Major Revisions

The manuscript presents an important and valuable research. However, I propose to follow the suggestions and comments of the reviewers to improve the manuscript substantially. The discussion requires improvement: the authors should highlight the significance of the results.

·

Basic reporting

Dear Authors,
After conducting a comprehensive review of the manuscript titled "Evaluation of COVID-19 severity and mortality association with ACE1 Alu 287 bp polymorphism and ACE1, ACE2, and TMPRSS2 expression in hospitalized patients " in the PeerJ with Submission ID 100658, I have identified several areas that require attention and improvement:

1. Title
It is better to change title name to “Genetic Determinants of COVID-19 Severity and Mortality: ACE1 Alu 287 bp Polymorphism and ACE1, ACE2, TMPRSS2 Expression in Hospitalized Patients”
2. In the Abstract section
• The Objective (lines 36-38) must be replaced by “This study aims to determine whether ACE1, ACE2, and TMPRSS2 mRNA expression levels, along with the ACE1 Alu 287 bp polymorphism (rs4646994), contribute to the severity and mortality of COVID-19”
• In the Methods section, replace the word “sample” with “swab” in lines 40 and 41
• In the Methods section, remove the sentence “Statistical analyses were performed using the R software environment (version 4.1.2)” from lines 45 and 46
• Keywords, Line 54-55, the word “genetic association” must be removed and replaced by polymorphism

3. In the “Introduction” section

• Lines 63-64, the sentence “Several clinical and epidemiological factors have been associated to COVID-19 severity in Brazil” must be rephrased to "Various clinical, genetic and epidemiological factors have been linked to COVID-19 severity worldwide”

• Lines 64-66, must be rephrased to “The degree of severity of COVID-19 disease, or vulnerability to SARS-CoV-2, depends on many factors, including genetic polymorphisms which studied in the following: transmembrane protease serine 2 (TMPRSS2), tumor necrosis factor-alpha (TNF-α), interferon-gamma (IFN-γ), and angiotensin-converting enzyme II” and add appropriate references

Experimental design

4. In the “Materials and Methods” section
• In the "Statistical analysis" section,
1. Please mention which type of normality tests (such as Kolmogorov-Smirnov, Shapiro-Wilk, or D'Agostino tests) were applied in the Statistical Analysis section. Did you attempt to transform the data before utilizing non-parametric tests?
2. In lines 134-135, state the p-value of the Hardy-Weinberg equilibrium test. Describe the method used to assess this equilibrium, and if equilibrium is not observed in your study, explain the approach taken to address this.
3. How was the sample size determined? Was G*Power software used for this determination?

Validity of the findings

5. In the "Results" section
• To enhance table comprehension without the need to refer back to the main text, it is essential to include a comprehensive description in the table footnote. For instance, provide details about the analysis method used for each entry in the table, and ensure that all abbreviations are fully expanded for reader clarity. This practice enables readers to grasp the information presented in the table independently, promoting a more seamless understanding of the data.

6. In “ Discussion” section
• The discussion lacks coverage of your results, overshadowing your significant findings. To enhance the discussion, compare your results with findings from other studies globally. Integrating such comparisons can enrich the discussion and highlight the broader implications of your study's results.
• Please include additional limitations of the study, such as the relatively small sample size and the absence of testing for linkage disequilibrium between genes in line 217-213.

Additional comments

7. The English writing in the manuscript requires editing. Additionally, there are superfluous commas and punctuation errors. Moreover, once an abbreviation has been mentioned, it should not be reiterated in detail later.

Reviewer 2 ·

Basic reporting

In the overall manuscript, the English language should be improved to ensure that an international audience can clearly understand your text.
I suggest you have a colleague who is proficient in English and familiar with the subject matter review your manuscript, or contact a professional editing service.
Some examples where the language could be improved include lines 61, 62 e.g. could be changed to:
The clinical manifestation of COVID-19 can be highly heterogenous with patients ranging from asymptomatic ...

63 can be changed to: Several clinical and epidemiological factors have been associated with COVID-19 ...

The current phrasing makes comprehension difficult.

Literature is well referenced and relevant.

Figures are relevant, high quality, well labelled and described.

Experimental design

Conclusions are well stated, linked to original research question and limited to supporting results.

Where are scripts/codes for the method?

I suggest that the authors make their work more computationally reproducible and provide the GitHub repository where they deposited the codes that were used to run the statistical analyses for this project.

Methods were described with sufficient detail but more information to replicate the statistical analysis needs to be provided.
It would be great to have a repository were the code for the computational methods for the statistical analysis in this study are deposited.

Validity of the findings

The underlying data have been provided. The study is limited by a few things including the vaccination status of the subjects as well as viral diversity impact.

The authors reported no significant association between ACE2 and TMPRSS2 gene expression and the need for mechanical ventilation or death. Although, previous studies have also noted that no ACE2 gene expression differences were found between those admitted to the intensive care unit and patients who were not (Akbari et al., 2020).

---

## Round 0.2 · accepted · Accept

There are a few minor suggestions of the reviewer to be considered in the process of submitting the final version. I ask that the authors attend to these while in production.

Reviewer 2 ·

Basic reporting

The authors did a good job of incorporating the comments from the previous version.
The present manuscript is well improved.

It will be good for reproducibility if the authors can include the code that was used for the statistical analyses that were conducted using the R software environment (version 4.1.2).

A few grammatical issues that I noted:
Line 72: Put a space between outcomes and the parenthesis.
and their polymorphisms with COVID-19 outcomes (Rossi et al.,

Line 211: Put a space after patients
higher in COVID-19 intensive care unit patients (Akbari et al., 2022)

Experimental design

Conclusions are well stated, and linked to original research question and limited to supporting results.

Validity of the findings

The underlying data have been provided. The study is limited by a few things including the vaccination status of the subjects as well as viral diversity impact.

The authors found no significant association between ACE2 and TMPRSS2 expression and the need for mechanical ventilation or death.